# The Strength in Axial Compression of Aluminum Alloy Tube Confined Concrete Columns with a Circular Hollow Section: Experimental Results

**Di Zhao [1], Jigang Zhang [1,*], Ling Lu [2], Haizhi Liang [1] and Zhehao Ma [1]**

[1] College of Civil Engineering, Qingdao University of Technology, Qingdao 266033, China; zhaodi2022@126.com (D.Z.); mr_liangok@163.com (H.L.); mazhehao@qut.edu.cn (Z.M.)

[2] Economic & Technology Research Institute, State Grid Shandong Electric Power Company, Jinan 250021, China; luling2018@126.com

[*] Correspondence: jigangzhang@126.com

**Abstract:** Steel tube confined concrete (STCC) stub columns have great strength and facilitate construction. In this study, the axial compressive strength of an aluminum alloy tube confined concrete column with (ATCC-CHS) and without (ATCC) a circular hollow section was tested in laboratory experiments. The influence of concrete strength, diameter–thickness ratio and the hollow rate on the failure mode, ultimate compressive strength, strain, stiffness, constraint effects and ductility was quantified. The experiments showed that local buckling failure could be effectively delayed when the outer aluminum tube did not directly bear axial load. Columns without a circular hollow section had bearing capacities approximately 20% higher than those with a circular hollow section, though their ductility was poorer. The ultimate strength tended to increase with decreases in the hollow rate and diameter–thickness ratio. It tended to increase with increasing concrete strength, though stronger concrete also reduced ductility. The bearing capacities of the columns were calculated according to several proposed formulas and compared with the experimental results, and the proposed Teng and Attard's formula appeared to be satisfactory.

**Keywords:** aluminum alloy tubes; confined concrete; columns with hollow section; axial compression; bearing capacity



## 1. Introduction

A steel tube confined concrete (STCC) stub column is constructed by pouring concrete into the tube. Breaking the tube at the ends allows any axial load to be borne only by the concrete core, as shown in Figure 1a. Gardener and Jacobson [1] first studied the axial compressive behavior of STCCs under different loads. The STCC concept was first proposed by Sakino et al. and Tomii et al. [2,3]. Later, various mechanical tests, theoretical analyses [4–7] and numerical simulations [8,9] were performed to study the strength and structural performance of STCCs. STCCs were found to have greater load-bearing capacities and better ductility than complete concrete-filled steel tubes (CFSTs) (see Figure 1b). The outer steel tube was disconnected at the joint, which could avoid or delay local buckling of the thin steel tube and improve the tube's tensile performance. The concrete was fully constrained, and brittle shear failure of the concrete was avoided. Moreover, the load conditions at the joint were greatly simplified, as shown in Figure 2.

Their high bearing capacity and ductility, convenient construction and excellent economic benefits have given STCCs broad application in composite structures [10,11]. Currently, they are mainly clad in carbon steel, but that has poor corrosion resistance, limiting the application of STCCs in severe environmental conditions, especially marine environments.

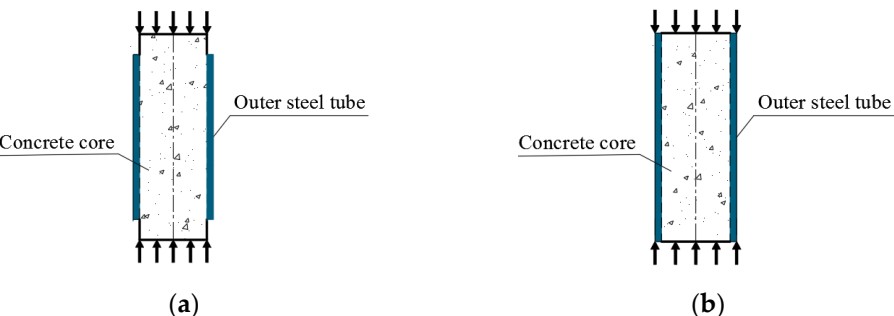

**Figure 1.** STCC and CFST columns. (**a**) STCC; (**b**) CFST.

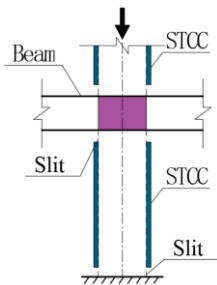

**Figure 2.** Schematic diagram of a beam–STCC column joint.

Aluminum alloy is now widely applied in high-rise buildings. Its advantages include its light weight, good strength, corrosion resistance, easy extrusion molding and low cost [12]. Aluminum has a low density and a low elastic modulus. Substituting aluminum for the steel in an STCC would not only improve its corrosion resistance, but also reduce its weight by 35% with the same material strength.

Aluminum tube-confined concrete (ATCC) is a new and relatively little-studied structural idea. Only one study using finite element simulation has simulated the axial compressive behavior of an ATCC column using 7075 high-strength aluminum [13]. A few studies have been carried out on the performance of concrete-filled aluminum tubes (CFAT). Axial compression tests of CFATs with square, rectangular and circular cross-sections showed that the failure modes are local buckling, cracking of the aluminum tube, and concrete crushing around the tube. Concrete with higher strength increases a CFAT's initial stiffness but reduces its ductility. Appropriate width-to-thickness ratios have been defined [14–16]. The design strength in the Australian/New Zealand standard [17] and the AA standard [18] was shown to be relatively conservative. A finite element simulation of a circular CFAT under an axial load using experimentally determined material properties predicted ultimate strength considering the strengthening effect of the compound section of the aluminum tube and the concrete [19]. Later, Wang et al. [20] analyzed the multiaxial stress state of the aluminum tube and the interaction between the tube and the concrete. They concluded that the method for calculating the nominal yield strength of a CFST proposed in the Chinese technical code for concrete-filled steel tubular structures [21] was also applicable to CFAT. The "unified theory" proposed by Zhong [22], Gong and Zha et al. [23,24] put forward an equation for calculating the combined strength of a CFAT under axial compression based on physical tests and finite element analysis. A group led by Patel [25] carried out numerical simulation of axial compression of a CFAT using a new confined concrete model and proposed an equation for a CFAT's ultimate axial strength based on the Liang–Fragomeni equation. A study led by Zeng et al. analyzed the buckling or oblique shear failure of a CFAT under axial compression and proposed an axial compressive strength equation based on a confinement effect coefficient [26]. Using a triaxial plastic damage constitutive model of concrete and an elastic-plastic constitutive model of aluminum alloy, Ding et al. [27] proposed an equation for estimating the ultimate strength of a column based on superposition.

Some have subsequently proposed pouring concrete between two concentric aluminum tubes to form a concrete-filled double aluminum alloy tube (CFDAT) with a hollow center. The inner tube increases the column's flexural stiffness while the overall weight is reduced. Zhou et al. [28] observed local buckling of the aluminum tube in axial compression and shear failure of the sandwiched concrete. They proposed a modified ultimate strength equation. Patel et al. [29] then performed a numerical analysis of the axial compression bearing capacity of a CFDAT and found that the AISC 360-16 and Eurocode 4 standards could overestimate or underestimate the strength. Their proposed model produced more accurate results. This method of aluminum usage reduces the weight by 22.5%.

So there have been studies showing that the CFAT and CFDAT configurations have advantages in terms of lighter weight, better corrosion resistance, better appearance and easier maintenance. However, compared with an STCC, the aluminum tube bears much of the axial load in a CFAT, which may lead to local buckling. That makes the configuration less suitable for high-load applications. Moreover, welding aluminum requires special processing, which limits the use of aluminum in composite structures.

To exploit the complementary advantages and to improve its material utilization, Han et al. [30] has designed A stainless steel–concrete–steel tube column and conducted a series of compressive tests. Wang et al. [31] subsequently studied the compressive behavior of such columns under internal and external pressure and Ye et al. [32] carried out axial compression experiments and simulations of the composite double-wall stainless steel–carbon steel concrete column. Wang et al. [33,34] studied the axial compression behavior of the stainless–concrete–high-strength steel composite. Ye et al. [35] studied the compressive behavior of CFAT stub columns with an inner carbon steel tube, and the results showed that increasing the size of the inner steel tube increased a column's strength and ductility, though using higher-strength concrete reduced the ductility. Work by Gao [36] proved the applicability of stainless-low-carbon steel-confined concrete columns in structural engineering due to their high axial compression strength and ductility.

In this research, a new aluminum alloy tube-confined concrete column with a circular section (termed an ATCC-CHS) was studied based on the CFDAT and STCC designs (Figure 3). As the outer aluminum tube is disconnected at the ends, the axial load is applied only to the inner steel tube and the concrete, while the concrete is constrained by both the inner and outer tubes. Compared with a conventional STCC, an ATCC-CHS was found to have similar bearing capacity, but better corrosion resistance, durability and economics. Axial compression tests were carried out on 24 ATCC-CHS and ATCC specimens to study the influence of various factors such as the radius–thickness ratio, hollow rate and the concrete's strength on their behavior in axial compression, failure modes, ultimate compressive strength, stiffness, ductility and strain characteristics. The constraint effects of ATCC-CHS and ATCC specimens were compared, and the axial ultimate compressive strength of the specimens was calculated.

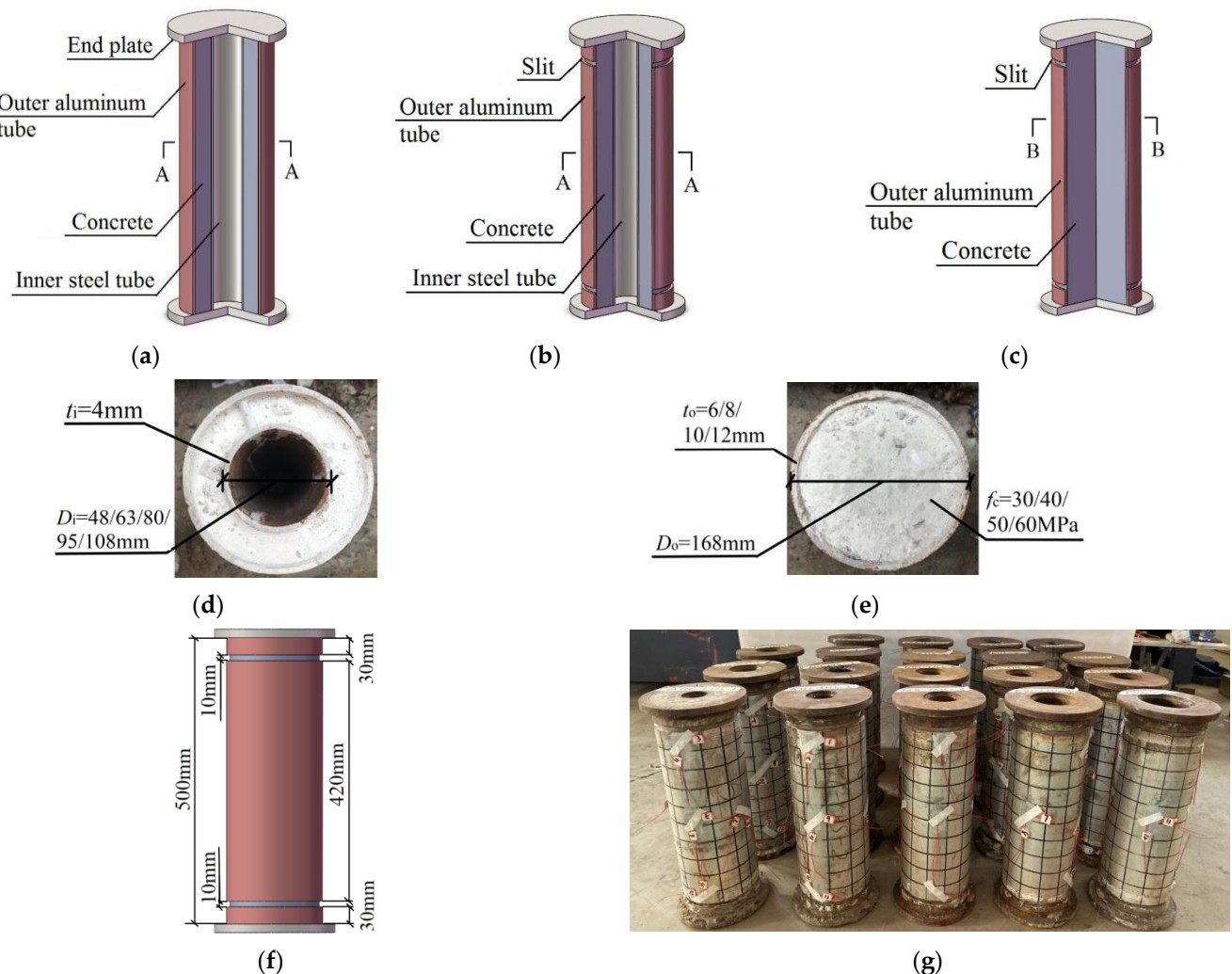

**Figure 3.** Specimen details.(**a**) CFAT-CHS columns; (**b**) ATCC-CHS columns; (**c**) ATCC columns; (**d**) A-A cross section; (**e**) B-B cross section; (**f**)specimen dimensions; (**g**) photos of specimens.

## 2. Test Program

### 2.1. Specimens

Twenty-four columns with a height $L$ of 500 mm and an external diameter $D_o$ of 168 mm were tested. To avoid end effects due to the short length and also buckling due to excess length, the specimens had a height–diameter ratio of 3.0. The main variables were the strength of the concrete (nominal cubic strengths $f_{cu}$ of 30, 40, 50 and 60 MPa), the diameter–thickness ratio ($D_o/t_o$ = 14–28), section hollow ratios ($\eta$ = 0.31–0.69), and confinement coefficients $\zeta$ = (0.97–2.21). See Table 1. Nineteen T6061 extruded seamless aluminum tubes were used as the outer tubes. Figure 3 shows the three design groups studied. Group A has four CFAT-CHS columns. Under axial load, the inner and outer tubes and the concrete were all subjected to compression. Group B was composed of 11 ATCC-CHS columns. A 10 mm slit was made in the outer aluminum tube approximately 100 mm from each end such that the outer tubes did not bear the axial load, and only the inner steel tube and the concrete were subjected to compression. Group C was composed of four ATCC columns. Their outer aluminum tubes were also cut, and only the concrete core was subjected to the axial compression. In addition, five STCC columns with Q235 seamless steel tubes comprised the control group. The ATCC-CHS specimens were labeled as AX-Y-Z, the CFAT-CHS specimens as CX-Y-Z and the ATCC specimens as AX-Y. Here, A represents ATCC, C represents CFAT, X is the outer aluminum wall thickness, Y is the concrete grade, and Z is the external diameter of the inner steel tube.

**Table 1.** Basic parameters of the specimen.

| Design | Specimen No. | $D_o \times t_o$ | $D_i \times t_i$ | $D_o/t_o$ | $D_i/t_i$ | $K \times 10^5$ | $\eta$ | $f_{cu}^{10}$ | $\zeta$ | Type of N-Δ Curve | $k\%$ | SI | SLI | $N_u$ | Material of the Outer Tube |
|---|---|---|---|---|---|---|---|---|---|---|---|---|---|---|---|
| a | C6-50-80 | 168 × 6 | 80 × 4 | 28 | - | 2.08 | - | 56.6 | 0.97 | A | - | - | - | 2198 | T6061 |
| | C8-50-80 | 168 × 8 | 80 × 4 | 21 | - | 2.17 | - | 56.6 | 1.36 | A | - | - | - | 2432 | T6061 |
| | C10-50-80 | 168×10 | 80 × 4 | 17 | - | 2.38 | - | 56.6 | 1.76 | B | 240 | - | - | 2779 | T6061 |
| | C12-50-80 | 168 × 12 | 80 × 4 | 14 | - | 2.80 | - | 56.6 | 2.21 | C | - | - | - | 3205 | T6061 |
| b | A6-50-48 | 168 × 6 | 48 × 4 | - | 12 | 2.14 | 0.31 | 56.6 | - | A | - | - | - | 2363 | T6061 |
| | A6-50-63 | 168 × 6 | 63 × 4 | - | 15.8 | 2.09 | 0.40 | 56.6 | - | A | - | - | - | 2257 | T6061 |
| | A6-50-80 | 168 × 6 | 80 × 4 | 28 | 20 | 1.68 | 0.51 | 56.6 | 0.97 | A | - | - | −3 | 2122 | T6061 |
| | A6-50-95 | 168 × 6 | 95 × 4 | - | 23.8 | 1.47 | 0.61 | 56.6 | - | A | - | - | - | 1982 | T6061 |
| | A6-50-108 | 168 × 6 | 108 × 4 | - | 27 | 1.28 | 0.69 | 56.6 | - | A | - | - | - | 1676 | T6061 |
| | A6-30-80 | 168 × 6 | 80 × 4 | - | - | 1.12 | - | 38.0 | 1.40 | B | - | - | - | 1807 | T6061 |
| | A6-40-80 | 168 × 6 | 80 × 4 | - | - | 1.51 | - | 48.3 | 1.12 | B | - | - | - | 1996 | T6061 |
| | A6-60-80 | 168 × 6 | 80 × 4 | - | - | 1.82 | - | 62.7 | 0.86 | A | - | - | - | 2277 | T6061 |
| | A8-50-80 | 168 × 8 | 80 × 4 | 21 | - | 1.73 | - | 56.6 | 1.36 | B | 262 | | −12 | 2138 | T6061 |
| | A10-50-80 | 168 × 10 | 80 × 4 | 17 | - | 1.79 | - | 56.6 | 1.76 | B | 339 | | −14 | 2396 | T6061 |
| | A12-50-80 | 168 × 12 | 80 × 4 | 14 | - | 1.79 | - | 56.6 | 2.21 | C | 437 | | −19 | 2603 | T6061 |
| | STCC-CHS-1 | 168 × 6 | 80 × 4 | 28 | - | 1.71 | - | 56.6 | 1.00 | - | - | - | - | 2242 | Q235 |
| c | A6-50 | 168 × 6 | - | 28 | - | 1.36 | - | 56.6 | 0.97 | A | 282 | 1.43 | 14 | 2319 | T6061 |
| | A8-50 | 168 × 8 | - | 21 | - | 1.50 | - | 56.6 | 1.36 | A | 343 | 1.46 | 18 | 2681 | T6061 |
| | A10-50 | 168 × 10 | - | 17 | - | 2.11 | - | 56.6 | 1.76 | C | 396 | 1.43 | 18 | 2927 | T6061 |
| | A12-50 | 168 × 12 | - | 14 | - | 2.94 | - | 56.6 | 2.21 | C | 482 | 1.50 | 25 | 3376 | T6061 |
| | STCC-1 | 168 × 6 | - | 28 | - | 2.36 | - | 56.6 | 0.97 | - | - | 1.24 | - | 2427 | Q235 |
| | STCC-2 | 168 × 8 | - | 21 | - | - | - | 56.6 | 1.36 | - | - | 1.22 | - | 2653 | Q235 |
| | STCC-3 | 168 × 10 | - | 17 | - | - | - | 56.6 | 1.76 | - | - | 1.20 | - | 3040 | Q235 |
| | STCC-4 | 168 × 12 | - | 14 | - | - | - | 56.6 | 2.21 | - | - | 1.21 | - | 3247 | Q235 |

Note: $D_o$, $t_o$, $D_i$ and $t_i$ are the diameters and thicknesses of the outer tube and the inner steel tube, respectively. $\eta$ is the hollow rate, $\eta = D_i/(D_o - 2t_o)$. $f_{cu}^{10}$ is the concrete's compressive strength in MPa, $\zeta$ is the confinement coefficient, $f_{ck}$ is the standard axial compressive strength of the concrete, $A_{ao}$ is the sectional area of the outer aluminum tube, and $A_{ce}$ is the nominal sectional area of the concrete, $A_{ce} = \pi (D_o - 2t_o) 2/4$.

To obtain the average yield strength $f_y$ ($f_{0.2}$), tensile strength fu, elastic modulus Eo and fracture elongation $\delta$ of the materials, three standard specimens were cut longitudinally from the outer aluminum alloy tube and the inner steel tube and tensile tests were carried out. The results are shown in Table 2. Since the elastic modulus of the aluminum alloy was smaller than that of steel, and there was no obvious yield point in the stress–strain curves, when the plastic strain was 0.2%, the corresponding non-proportional ultimate strength $f_{0.2}$ was taken as the yield strength of the aluminum alloy. Then, $150 \times 150 \times 150$ mm and $150 \times 150 \times 300$ mm concrete specimens were prepared with four grades (C30, C40, C50 or C60). After 12 days of curing, their strengths were measured according to China's GB/T50081-2002 standard [37]. The compressive strength fcu and axial compressive strength $f_{ck}$ data are presented in Table 3.

**Table 2.** Mechanical properties of the steel tubes.

| Steel Type | Yield Strength $f_y$ ($f_{0.2}$)/MPa | Tensile Strength $f_u$/MPa | Elastic Modulus $E_s$/GPa | Elongation $\delta$/% |
|---|---|---|---|---|
| Q235 | 357 | 394 | 182 | 46.88 |
| T6061 | 263 | 288 | 71 | 22.8 |

**Table 3.** Compressive strength and axial compressive strength of the concrete.

| Concrete Grade | Cube Compressive Strength $f_{cu}$/MPa | Axial Compressive Strength $f_{ck}$/MPa |
|---|---|---|
| C30 | 41 | 29.8 |
| C40 | 52.3 | 37.6 |
| C50 | 61.3 | 43 |
| C60 | 67.0 | 48.9 |

## 2.2. Test Setup

The loading device for the axial compression testing is shown in Figure 4a. It was a 5000 kN Shanghai Hualong YJW-5000 electro-hydraulic servo compression-shear test machine. The loading protocol was as follows: First, the specimens were pre-loaded under 10% of the estimated ultimate strength. Stepwise loading was then applied in increments of

approximately 1/10th of the expected ultimate load which lasted for 2 min. After reaching 80% of the expected ultimate load, the specimen was loaded under displacement control at 1 mm/min. When the load reached or was close to the expected ultimate load, the loading rate was changed to 0.5 mm/min. The loading was stopped when the bearing capacity dropped to 85% of the peak load or in the case of severe deformation.

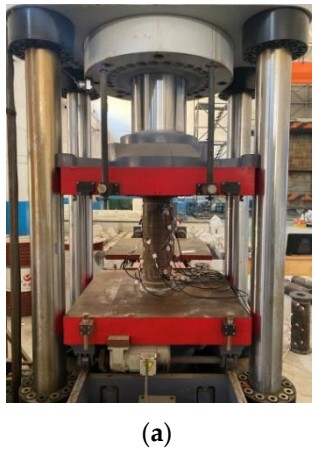
(a)

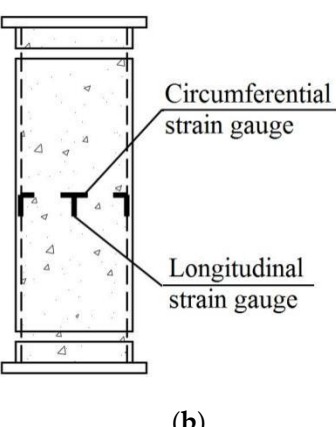
(b)

**Figure 4.** Testing setup and placement of the measuring points. (**a**) Axial compression test of columns. (**b**) The arrangement of strain gauges.

### 2.3. Instrumentation

The following items were measured during the tests: the axial load, vertical displacement, and longitudinal and circumferential strain at the middle of the outer aluminum tube. Compressive deformation and changes in the slits at two ends of the outer tube were also observed and recorded. The layout of the measurement points is shown in Figure 4b. Specifically, in the middle of the outer aluminum tube, four groups of resistance strain gauges were placed 90° apart so that the two gauges of each group were perpendicular. There were thus eight strain gauges. The strain gauge data were collected using a DH3816 static data acquisition module. The axial load was recorded by the test machine, and the axial displacement was measured using pull-wire displacement sensors.

## 3. Results and Analysis

### 3.1. Failure Modes

Some of the $N$-$\Delta$ curves had no descending stage, so no large deformation occurred under the peak load and it is unreasonable to consider the peak load as the ultimate load. In those cases, $N_u$ was defined as the load corresponding to a measured longitudinal strain $\varepsilon_v$ of 5000 $\mu\varepsilon$ [38].

Typical failure modes of the CFAT-CHS specimens were circumferential buckling at the top of the outer aluminum tube, longitudinal microcracking, and localized concrete crushing, as shown in Figure 5a. In the elastic deformation stage, there was no obvious deformation. When the load increased to 0.6–0.7 $N_u$, localized circumferential buckling failure occurred in the upper part of the outer aluminum tube approximately 50 mm from the end plate. It was similar to the axial crushing failure of an aluminum tube. When the loading value was close to $N_u$, the outer aluminum tube began the necking stage due to strain strengthening of the aluminum alloy material, and there were multiple axial microcracks at both sides of the buckled ring. Loud cracking was heard. The concrete was broken and raised, and the inner steel tube was slightly buckled.

The failure processes of the ATCC specimens were similar to that of the ATCC-CHSs. There were two failure modes depending on the thickness of the outer wall. Failure mode I: longitudinal cracking in the outer aluminum tube, shear failure in the concrete, and bending failure of the inner steel tube (Figure 5b,c). The failures were brittle ones. Failure mode II: local buckling of the outer aluminum tube, local crushing failure of the concrete,

and local buckling of the inner steel tube (Figure 5d,e). Those were ductile failures. In the elastic deformation stage, there was no obvious deformation of the specimens. When the load reached 0.6–0.7 $N_\mathrm{u}$, the slit at the top of the aluminum tube narrowed, the concrete at the cracks was crushed, and there was slight buckling in the middle part of the aluminum tube. When the loading was close to Nu, strain strengthening of the thin aluminum alloy tube ($t_\mathrm{o} \leq 8$ mm) caused necking, and a crack formed running through the axial direction. This was accompanied by loud cracking sounds. Meanwhile, diagonal cracks with a shear angle of 45–60° formed in the concrete at the aluminum tube buckling site, indicating brittle failure. With further load increases, the crack in the outer aluminum tube widened, the confinement degraded, and a large amount of concrete was crushed and fell off. When the load increased to 1.05 $N_\mathrm{u}$ the thicker aluminum alloy tube ($t_\mathrm{o} \geq 10$ mm) entered the plastic stage while the bearing capacity was still increasing slowly. There was local buckling at both ends of the aluminum tube, where local crushing failure of the concrete and diamond-like failure of the inner steel tube were observed. The bearing capacity remained high even after yielding. In addition, there was pyramidal local compression failure in the concrete at the ends of the ATCC samples. Compared with the aluminum tube in the CFAT-CHS specimens, the aluminum tube in the ATCC and ATCC-CHS specimens with the same confinement coefficient effectively delayed local buckling, since the outer aluminum tube was not directly loaded.

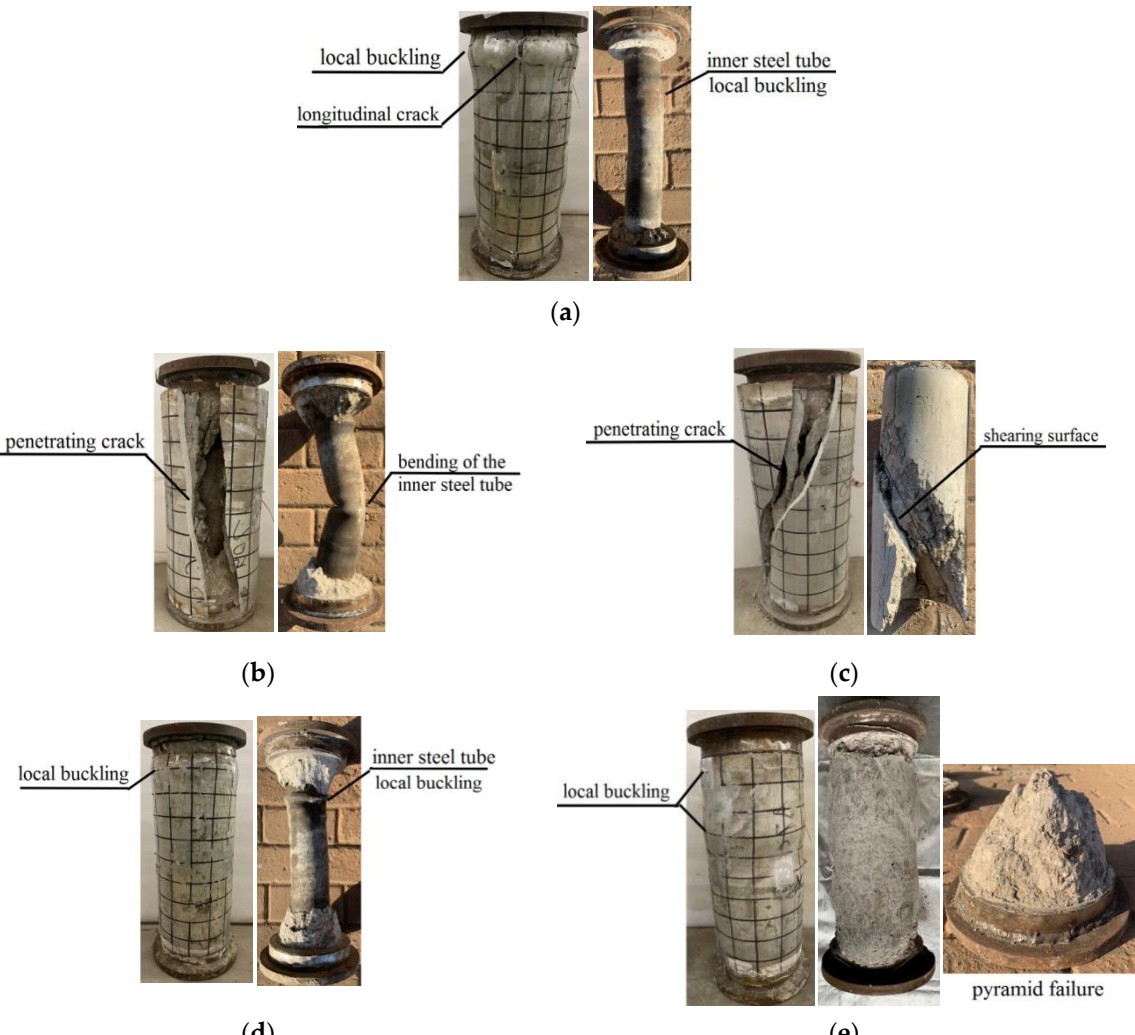

**Figure 5.** Failure modes. (**a**) CFAT-CHS failure, (**b**) ATCC-CHS failure mode I, (**c**) ATCC failure mode I, (**d**) ATCC-CHS failure mode II, and (**e**) ATCC failure mode II.

Both the STCC-CHS-1 and STCC-1 specimens had ductile failures with multiple localized areas of circumferential buckling. The thin-walled ATCCs had poor confinement, leading to brittle failure. Moreover, since the column-end constraint was not considered in the specimen design, there were uneven loads on the column, causing column-end failure. In addition, although the axial shortening of the different materials was similar, and although the thick-walled ATCC specimens showed ductile failure similar to that of the STCCs, the buckling deformation of the aluminum tubes was smoother than that of the carbon steel tubes. This was mainly because of their greater plasticity and strain hardening of the carbon steel. When there was steel tube buckling, the tube imposed effective lateral constraint on the internal concrete core, which allowed the load to increase continuously before any new local buckling occurred.

### 3.2. Comparison of Axial Loads

The axial load (*N*) and displacement (Δ) data are summarized in Figure 6. The *N*-Δ curves of all three types of specimens show ductile characteristics. At that point, there was already plastic deformation. Table 1 specifies all the ultimate loads. Figure 7 displays the axial compression stiffnesses K and tangent stiffnesses calculated at 0.4 $N_u$ [39].

The influence of the diameter–thickness ratio on the load–displacement curve of the ATCC-CHS specimens is shown in Figure 6a,e. Within the range tested in this study ($D_o/t_o$ = 14–28), Nu increased linearly with decreases in the diameter–thickness ratio of the outer tube. When the ratio decreased from 28 to 14, the ultimate load increased by 22.6%, and the axial compression stiffness increased by 6.5%. Since the inner steel tube was directly subjected to longitudinal loading, when the diameter–thickness ratio of the inner steel tube decreased from 27 to 12, the ultimate strength of the specimen increased by 41.0%. Therefore, the diameter–thickness ratio of the inner steel tube had more influence on the performance of the column. In addition, for an ATCC specimen the ultimate load increased by 45.6% due to the changing diameter–thickness ratio, indicating that the constraint of the outer aluminum tube plays a more significant role in the compressive strength of the concrete.

The influence of hollow rate on the load–displacement curves of the ATCC-CHS specimens is shown in Figure 6b. Within the range of *η* < 0.69, the ultimate load and the latter-stage bearing capacity of the ATCC-CHS specimens were higher than those of the ATCC specimens when the hollow rate was less than 0.4. Since the concrete core was replaced by the inner steel tube at a certain hollow rate, the overall compressive strength and ductility of the specimen increased. When the hollow rate decreased from 0.69 to 0.31, the bearing capacity of the specimen increased by 41.0%, and the axial compression stiffness increased by 67%.

The influence of concrete strength on the load–displacement curves of the ATCC-CHS specimens is shown in Figure 6c. Columns with C30 and C40 concrete showed good ductility, and there was no loss of bearing capacity when the peak load was reached. With the stronger concrete, the elastic stiffness and the ultimate strength of the ATCC-CHS columns increased. The ductility, however, decreased. Compared with the A6-30-80 specimen, the ultimate strength of the specimens increased by 10.5%, 17.4% and 26.0% when the concrete strength increased by 33%, 67% and 100%, respectively.

The load–displacement curves of the specimens with aluminum alloy and Q235 carbon steel outer tubes are shown in Figure 6d. In mode b, the peak load and axial compressive stiffness of the specimens confined in an aluminum alloy tube were 5.3% and 1.2% lower than those of similar specimens confined in a carbon steel tube, respectively. In mode c, the peak load and the axial compressive stiffness were 4.7% and 42.3% lower. Moreover, the ductility and the latter-stage ultimate strength of the ATCC specimens were poorer as well. This is because carbon steel has an obvious strengthening stage, which provides stronger constraint to the concrete core.

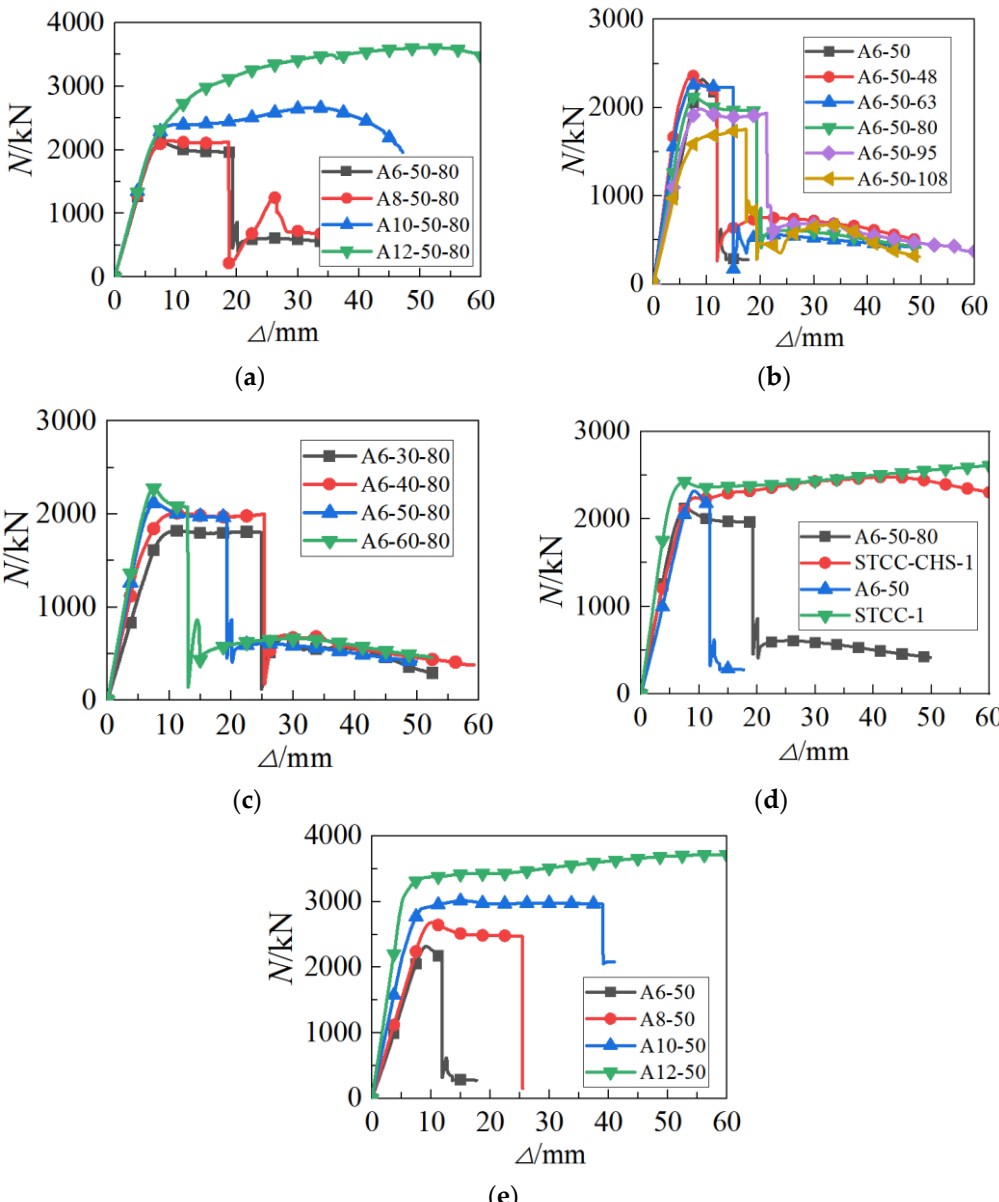

**Figure 6.** *N*-Δ relationships. (**a**) Diameter–thickness ratio, (**b**) hollow rate, (**c**) concrete strength, (**d**) material of the outer tube, and (**e**) ATCC.

In summary, the bearing capacity of an ATCC-CHS can be increased by reducing the diameter–thickness ratio and the hollow rate or by using stronger concrete. Within a certain range, increasing the wall thickness of the outer aluminum tube is the most effective way to improve a column's ultimate strength. A thick outer wall provides good constraint to the concrete core, preventing sudden brittle failure. Additionally, the strength and ductility can continue to increase after the peak load. The stiffness is proportional to the wall thickness of the aluminum tube and the concrete's strength and negatively related to the hollow rate. The confinement coefficient affected the stiffness of the ATCC specimens more than the ATCC-CHS specimens.

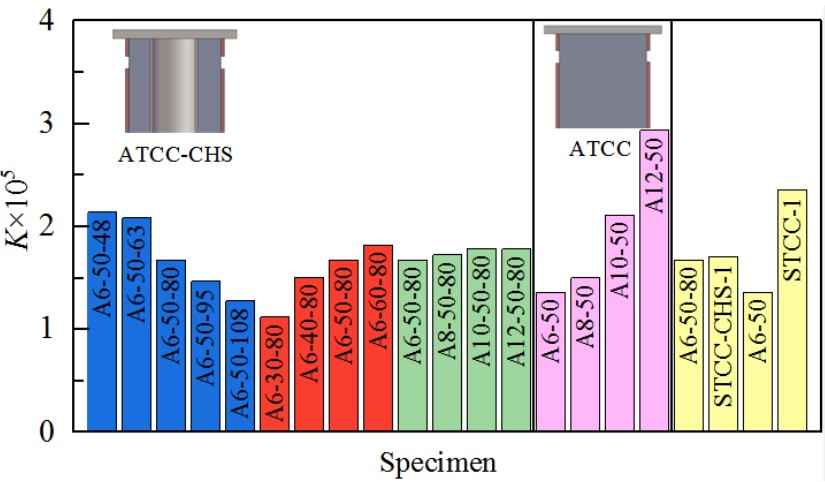

**Figure 7.** Stiffness comparison.

In order to study the influence of the outer aluminum tube's confinement coefficient on the axial compressive strength, the bearing capacities of ATCC-CHS and ATCC specimens were compared with that of plain concrete columns with the same geometry and material properties. The strength coefficient $k$ is defined as

$$k = \frac{N_{uATCC-CHS} - A_{si}f_{yi}}{A_c f_{ck}} \tag{1}$$

or

$$k = \frac{N_{uATCC}}{A_c f_{ck}} \tag{2}$$

where $N_{u\text{-}ATCC\text{-}CHS}$ is the measured ultimate strength of the ATCC-CHS specimen, $N_{u\text{-}ATCC}$ is the measured ultimate strength of the ATCC specimen, $A_{si}$ is the cross-sectional area of the inner steel tube, $f_{yi}$ is the yield strength of the inner steel tube, and $f_{ck}$ is the standard value of the concrete's compressive strength [40]. China's Code for the Design of Concrete Structures specifies that $f_{ck} = 0.88 \times 0.76 f_{cu}$. The bearing capacities of the ATCC and ATCC-CHS columns both increased significantly ($k = 482\%$ for specimen A12-50, $k = 437\%$ for specimen A12-50-80). The specimens' $k$ values are shown in Table 1. Figure 8 shows the $k$–$\zeta$ relationship curves. The nominal confinement factor is defined as follows,

$$\zeta = A_{ao} \cdot f_{0.2} / A_{ce} \cdot f_{ck} \tag{3}$$

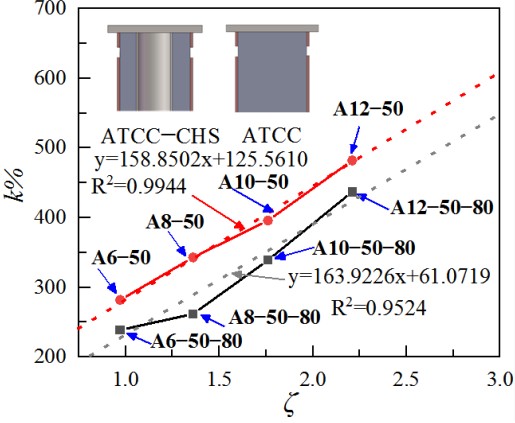

**Figure 8.** $k$–$\zeta$ relationship.

They show that $k$ increased with increases in the confinement coefficient $\zeta$ within the range $\zeta = 0.97$–$2.21$. This is because the concrete core has high compressive strength due to the lateral constraint of the aluminum tube, and the stronger the constraint, the greater the concrete's strength. However, the increased strength is not due entirely to the constraint. The outer aluminum tube bears a certain load transferred by friction at the contact surface. With the same confinement coefficient, the increase in bearing capacity of an ATCC is slightly greater than that of an ATCC-CHS.

The strength indexes *SI* (Equation (4)) of the ATCCs and CFATs were also compared. The ATCC data are shown in red dots in Figure 9, with the CFAT data in black dots [26,41]. The *SI*s of the ATCCs remained stable at approximately 1.5, increasing with the confinement coefficient. For $\zeta < 1.29$, the ultimate strength of the ATCCs was slightly lower than that of the CFATs under the same conditions. When $\zeta > 1.29$, it was higher.

$$SI = \frac{N_{\text{uATCC}}}{A_c f_{ck} + A_{ao} f_{0.2}} \quad (4)$$

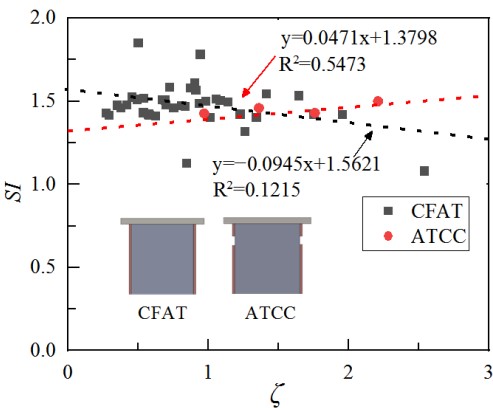

**Figure 9.** *SI*–$\zeta$ relationship.

To evaluate the influence of the confinement coefficient of the aluminum tube on the axial ultimate compressive strength, a bearing capacity coefficient, *SLI*, was defined as

$$SLI = \frac{N_{\text{uATCC–CHS}} - N_{\text{uCFAT–CHS}}}{N_{\text{uCFAT–CHS}}} \quad (5)$$

or

$$SLI = \frac{N_{\text{uATCC}} - N_{\text{uCFAT}}}{N_{\text{uCFAT}}} \quad (6)$$

where $N_{\text{u-CFAT-CHS}}$ is the measured ultimate strength of a CFAT-CHS specimen and $N_{\text{u-CFAT}}$ is the ultimate strength of a CFAT specimen calculated using the equation proposed in Hu and Zeng's work [26,41]. Figure 10 shows the *SLI*–$\zeta$ relationship. It shows that the bearing capacity coefficient, *SLI*, of an ATCC-CHS decreases as $\zeta$ increases (range: 3–19%) with the same hollow steel tube geometry. The bearing capacity coefficient, *SLI*, of an ATCC increases with $\zeta$ (range: 14–25%). The difference is due to different structures of the three specimens. The inner and outer steel tubes of a CFAT-CHS bear the longitudinal load jointly, so the axial load increment corresponding to a unit of deformation increases, leading to an increase in initial stiffness and rapid strengthening. The slit creates a weak section in an ATCC-CHS, so the inner steel tube buckles locally under load. The concrete at the slit does not have interior support and is in uniaxial compression. For specimens with a large hollow rate, the axial load on the inner steel tube is unable to offset the bearing capacity loss at the slit, thus the bearing capacity decreases. Under axial load, the internal solid concrete of an ATCC is still under triaxial constraint.

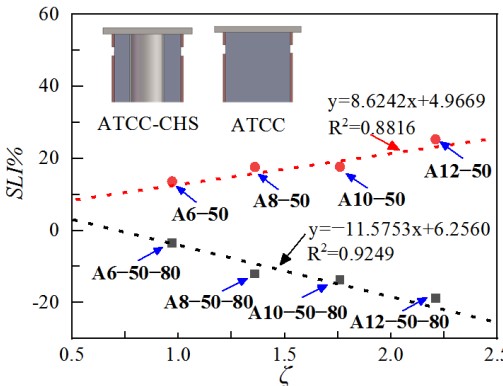

**Figure 10.** *SLI–ζ* relationships.

The axial ultimate strengths of the ATCC and STCC specimens differed due to the different constraint from the outer tube. Figure 11 compares the test results of similar ATCC and STCC specimens. The *SLI–ζ* relationship is clearly different. At the same ζ value, the *SI* value of an ATCC specimen is larger than that of an STCC specimen because the bond between carbon steel and concrete is stronger than that between aluminum and concrete. That provides greater longitudinal friction.

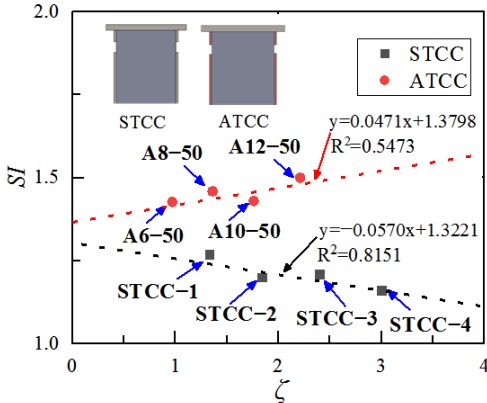

**Figure 11.** *SI–ζ* relationships.

### 3.3. Deformation under Axial Load

The typical axial load (*N*) and average strain ($\bar{\varepsilon}$) curves obtained in the tests are shown in Figure 12. $N_{max}$ is the ultimate load, and the *N*–$\bar{\varepsilon}$ curve changes from Type A to Type B and then to Type C, with increases in the confinement coefficient ζ. The curve types for different specimens are shown in Table 1. Type A is typical for conventional CFSTs. Such strain softening affects many metal materials when the deformation exceeds a certain threshold, and the failure mode is then brittle failure. In these experiments, the following ATCC specimens produced Type A curves: C6-50-80 (ζ = 0.97), C8-50-80 (ζ = 1.36), A6-50-80 (ζ = 0.97), A6-60-80 (ζ = 0.86), A6-50 (ζ = 0.97), and A8-50 (ζ = 1.36). Among them, A6-50-80 and A6-50 were relatively thin, and A6-60-80 had high-strength concrete. With a Type A curve, the load decreases with increases in the axial deformation after the peak load (point #1), but it is still within 80 to 90% of the ultimate load when reaching point #2. The load increases to point #3 at the end of the test due to strain strengthening of the aluminum alloy. The test results showed that the load value at point #3 was generally greater than that at point #1.

Type B *N*-Δ curves were observed with specimens C10-50-80 (ζ = 1.76), A6-30-80 (ζ = 1.41), A6-40-80 (ζ = 1.41), A10-50-80 (ζ = 1.36), and A10-50-80 (ζ = 1.76). The Type B curve specimens were relatively stable at the 1′–2′ section compared to specimens with a Type A curve, while the other sections were similar. The load stabilized after reaching the

peak load (point #1′). As the test continued, the load increased to point #3′, which could also be attributed to strain strengthening of the aluminum alloy.

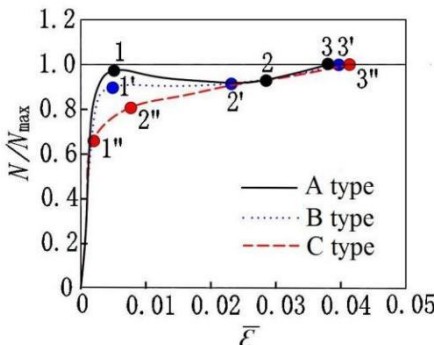

**Figure 12.** The three types of typical axial load (*N*) and average axial strain ($\bar{\varepsilon}$) curves.

In the Type C curves, there was a strengthening stage. The greater the $\zeta$ value, the more significant the strengthening. Type C curves were observed with specimens C12-50-80 ($\zeta$ = 2.21), A12-50-80 ($\zeta$ = 2.21), A10-50 ($\zeta$ = 1.76), and A12-50 ($\zeta$ = 2.21). The curve includes an initial linear elastic stage and a later linear plastic stage. The curve gradually flattens between #1″and #2″, which is mainly attributable to local buckling of the aluminum tube and non-linear response of the concrete. After point 2″, the $N$-$\bar{\varepsilon}$ curve increases linearly, and the slope is much smaller than in the 0–1″ section. The peak load was reached at the end of the test (point #3″) due to confinement by the aluminum after plastic deformation of the concrete.

In summary, the critical value of $\zeta$ for the ATCC-CHS specimens (Types A and B) was 0.97, and that for CFAT-CHSs was 1.36 (Figure 13). ATCC-CHSs have better deformation resistance at smaller $\zeta$ values.

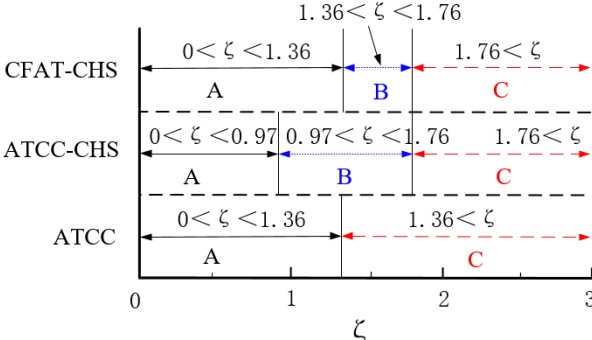

**Figure 13.** Curve type versus ξ relationships.

*3.4. Strain*

Figure 14 shows the relationship between the measured load of the three types of the specimens and the transverse $\varepsilon_h$ and longitudinal $\varepsilon_v$ strain at the middle section. The compressive strain is negative, and the tensile strain is positive. Before a specimen reached its ultimate strength, the longitudinal and transverse strains of the aluminum tube had already reached the yield strain (2432 $\mu\varepsilon$), suggesting that the strength of the aluminum tube was fully utilized. With the same geometry and material properties, the axial compression stiffness and bearing capacity of the ATCC specimens were higher than those of CFAT-CHS and ATCC-CHS specimens. Generally, the declining section of the load–longitudinal strain curve for CFAT-CHSs was smooth and stable, showing the best deformation performance. That was followed by the ATCC-CHSs and ATCCs.

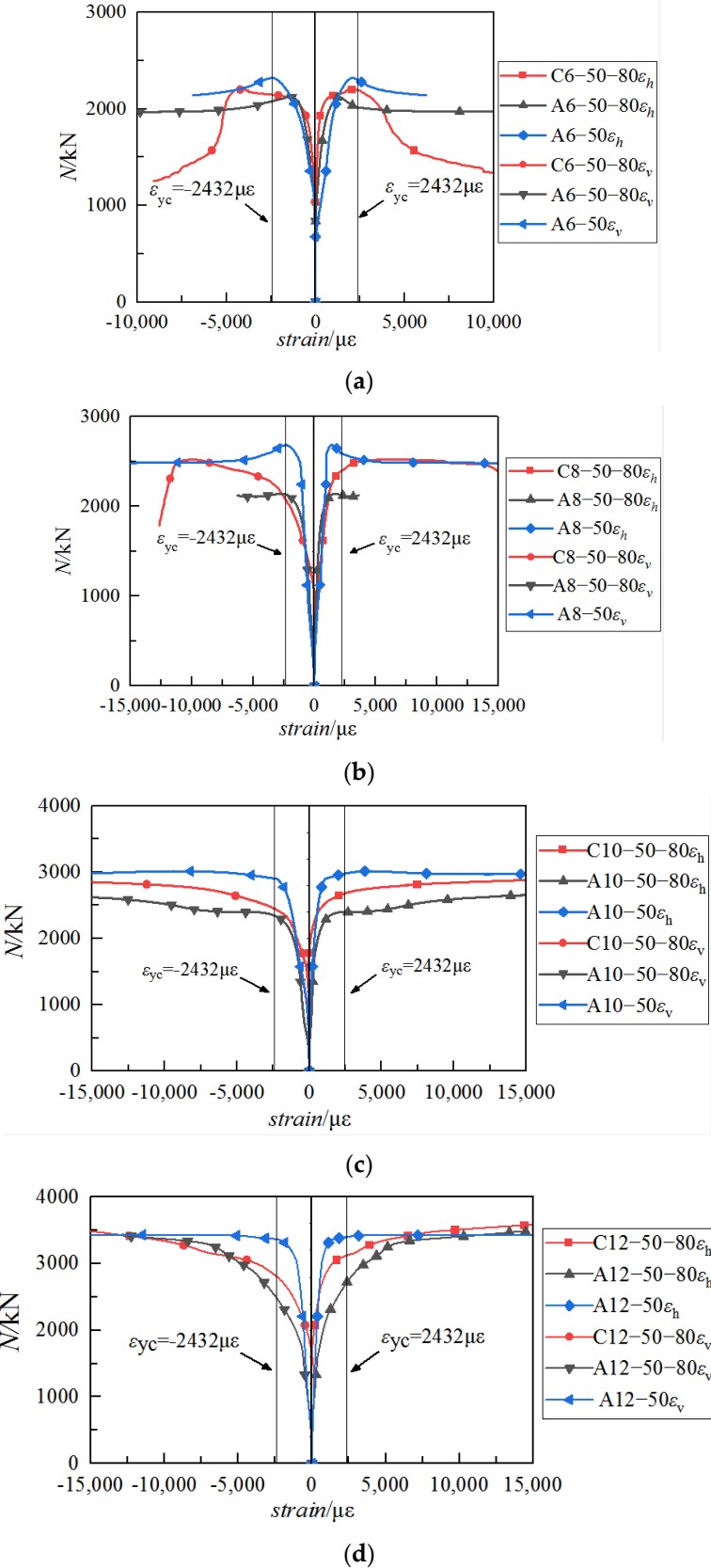

**Figure 14.** *N* and the middle section strain. (**a**) $D_o/t_o$ = 28, (**b**) $D_o/t_o$ = 21, (**c**) $D_o/t_o$ = 17, and (**d**) $D_o/t_o$ = 14.

Figure 15 shows the typical $N/N_u$-$\varepsilon_v/\varepsilon_h$ relationship curve of specimen A6-50-80. It shows that the longitudinal and circumferential strain of the aluminum tube increased linearly before the peak load was reached, and $\varepsilon_v$ increased faster than $\varepsilon_h$. Plus, the ratio between the transverse and longitudinal strains was larger than aluminum's Poisson's ratio ($\mu_s = 0.33$), indicating strong confinement. After the peak load, the longitudinal and transverse strains started to increase non-linearly. On average, the ratios of the CFAT-CHSs were greater than 2.1, and the ratios of the ATCC-CHSs and ATCCs averaged 1.5 and 1.1, respectively. This indicates that there was substantial deformation of the aluminum tube. Due to the expansion of the concrete, the circumferential strain increased rapidly and surpassed the longitudinal strain in the middle of the steel tube. Therefore, the circumferential strain was mainly loaded on the aluminum tube.

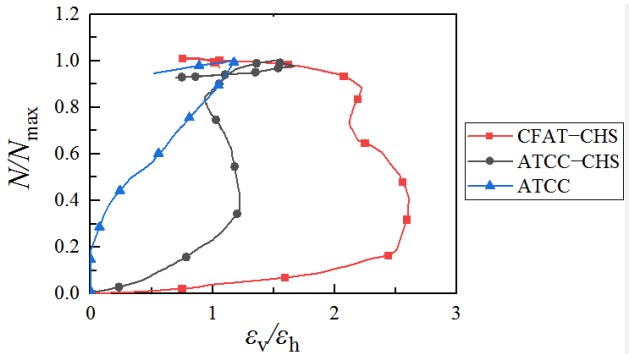

**Figure 15.** $N/N_{max}$-$\varepsilon_v/\varepsilon_h$ relationships.

Figure 16 presents the relationship between $\varepsilon_h$ and $\varepsilon_v$ to the ultimate load and the diameter–thickness ratio. With the same diameter–thickness ratio, there was little difference in the $\varepsilon_h$ and $\varepsilon_v$ values of the ATCC-CHSs and ATCCs. Yet both were significantly smaller than for CFAT-CHSs. This resulted from the different compression mechanisms provided by the outer aluminum tube. When the aluminum tube and the concrete were jointly loaded, the aluminum tube was directly subjected to longitudinal loading, so its confinement of the concrete was poor. However, when the load was only on the concrete and the inner steel tube, the outer aluminum tube loaded only axially by the friction force, resulting in little longitudinal strain and better confinement.

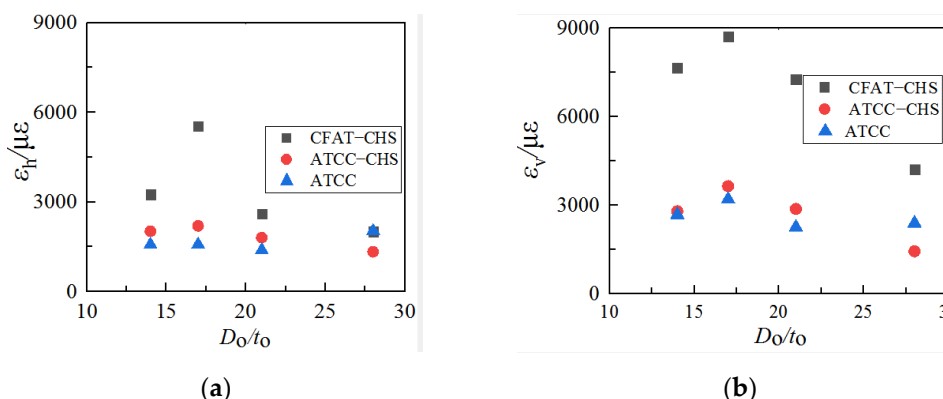

(a)  (b)

**Figure 16.** Relationship curve between $\varepsilon_h$ and $\varepsilon_v$ and the parameter-thickness ratio. (a) $\varepsilon_h$–$D_o/t_o$ relationships, (b) $\varepsilon_v$–$D_o/t_o$ relationships.

## 4. Predicting Axial Bearing Capacity

### 4.1. Axial Bearing Capacity of a CFAT-CHS

A CFAT-CHS can be regarded as a CFST-CHS with a lower modulus of elasticity. Its axial bearing capacity can be divided into the combined strength of the outer aluminum

tube and the concrete and that of the inner steel tube. According to Huang [42], when the hollow rate $\eta \leq 0.75$, the following simplified formula adequately predicts the axial bearing capacity of a CFST-CHS under condition a.

$$N_{c-\text{CFAT}-\text{CHS}} = A_{aco} \cdot f_{acy} + A_{si} \cdot f_{yi} \tag{7}$$

Here, the cross-sectional area of the outer aluminum tube and the sandwiched concrete together is calculated as $A_{\text{aco}} = A_{\text{ao}} + A_\text{c}$, the concrete strength is $f_{\text{acy}} = C_1\,\eta^2\,f_{0.2} + C_2$ $(1.14 + 1.02\,\zeta)\,f_{\text{co}}$, where $C_1 = \alpha/(1 + \alpha)$ and $C2 = (1 + \alpha_\text{n})/(1 + \alpha)$. The aluminum content is $\alpha = A_{\text{ao}}/A_\text{c}$, the nominal aluminum content is $\alpha_\text{n} = A_{\text{ao}}/A_{\text{ce}}$, and $A_\text{c}$ is the cross-sectional area of the concrete.

The calculated results show that the mean of $N_\text{u}/N_\text{c}$-CFAT-CHS is 1.05 and its standard deviation is 0.02. That agrees well with the experimental results, indicating that the test setup and measurements for the ATCC-CHS and ATCC specimens were reliable.

### 4.2. Axial Bearing Capacity of an ATCC

The bearing capacity of STCC has been extensively studied. The axial compressive strength fcc of a confined concrete column under triaxial compression is a function of the uniaxial compressive strength fc and the circumferential effective confining stress $f_1$. In this study, the calculation models proposed by Mander [43], Xiao [44], Teng [45] and Attard [46] were used (Table 4). For STCC columns with a circular section, the steel pipe resists transverse deformation of the concrete core providing lateral confining stress and axial compression. The concrete is thus in a triaxial stress state. According to Yu [11], the axial stress on a circular STCC at the steel's yield point is much lower than the circumferential stress. Therefore, the longitudinal stress on the steel tube was taken as zero, and the transverse stress was the yield strength of the steel tube ($\sigma_1 = 0$, $\sigma_\text{h} = f_{\text{yo}}$). $f_1$ can be calculated as

$$f_1 = 2t_\text{o} f_{\text{yo}}/(D_\text{o} - 2t_\text{o}) \tag{8}$$

**Table 4.** Proposals for calculating the strength of confined concrete columns.

| Model | Equation |
|:---:|:---:|
| Mander [43] | $f_{\text{cc}} = f_{\text{co}}\left[-1.254 + 2.254\sqrt{1 + 7.94\frac{f_1}{f_{\text{co}}}} - 2\frac{f_1}{f_{\text{co}}}\right]$ |
| Xiao [44] | $f_{\text{cc}} = f_{\text{co}}\left[1 + 3.24\left(\frac{f_1}{f_{\text{co}}}\right)^{0.8}\right]$ |
| Teng [45] | $f_{\text{cc}} = f_{\text{co}}\left(1 + 3.5\frac{f_1}{f_{\text{co}}}\right)$ |
| Attard [46] | $f_{\text{cc}} = f_{\text{co}}\left(\frac{f_1}{0.558\sqrt{f_{\text{co}}}} + 1\right)^{1.25[1+0.062\frac{f_1}{f_{\text{co}}}](f_{\text{co}})^{-0.21}}$ |

To calculate an ATCC's bearing capacity, it is necessary to first determine the load-bearing component. According to Gan [47], because of the end slits, the outer aluminum tube can be assumed to bear no direct axial load. Thus, any longitudinal stress on the outer aluminum tube is not considered in the calculation. It is converted into a circumferential stress on the concrete. The axial compressive strength of an ATCC is then calculated as

$$N_{c-\text{ATCC}} = A_\text{c} \cdot f_{\text{cc}} \tag{9}$$

The results of the Xiao, Teng and Attard models were close to the experimental results. Specifically, the results of the Xiao and Teng models were slightly smaller and the Attard model's predictions were slightly greater than the experimental results. Teng's formula came closest, with a coefficient of variation of 0.01. Thus, Teng's formula can best predict the Nu of an ATCC.

### 4.3. Axial Bearing Capacity of an ATCC-CHS

For a given hollow rate, the non-proportional ultimate strength ($f_{0.2}$), concrete strength ($f_{ck}$), and nominal aluminum content ($\alpha_n$) were found to be the main factors determining the strength of an ATCC-CHS. Their influences on the strength were similar to that on the strength of an STCC with a hollow section. In an ATCC-CHS, any axial load acts directly on the inner tube and the concrete. The outer aluminum tube simply provides circumferential confinement. Based on Equations (4) and (6), an ATCC-CHS's bearing capacity can be predicted as

$$N_{c-ATCC-CHS} = A_c f_{cc} + A_{si} f_{yi} \tag{10}$$

Table 5 shows the calculated bearing capacities of the ATCC-CHS specimens using the various proposed formulas. The calculated values are generally consistent with those measured (Figure 17). In particular, the Mander, Xiao, and Teng formulas are conservative, while the Attard model predicts greater strength. Thay Attard prediction was the closest to the experimental results. The average value from the Attard formula was 0.97 of the observed value with a coefficient of variation of 0.03. The average values of the Xiao and Teng models were 1.04 and 1.06, respectively. The error from Mander's equation grew with increases in the confinement coefficient. In common engineering practice where the confinement coefficient of an STCC is $\zeta < 2$, the Mander model effectively predicts the bearing capacity of an ATCC-CHS.

**Table 5.** Ultimate strength of the tested specimens.

| Specimen No. | $N_u$ | $N_{c-Mander}$ | $N_u/N_{c-Mander}$ | $N_{c-xiao}$ | $N_u/N_{c-xiao}$ | $N_{c-Teng}$ | $N_u/N_{c-Teng}$ | $N_{c-Attard}$ | $N_u/N_{c-Attard}$ |
|---|---|---|---|---|---|---|---|---|---|
| A6-50-48 | 2363 | 2212 | 1.07 | 2259 | 1.05 | 2165 | 1.09 | 2421 | 0.98 |
| A6-50-63 | 2257 | 2127 | 1.06 | 2170 | 1.04 | 2084 | 1.08 | 2321 | 0.97 |
| A6-50-80 | 2122 | 1981 | 1.07 | 2019 | 1.05 | 1943 | 1.09 | 2152 | 0.99 |
| A6-50-95 | 1982 | 1809 | 1.10 | 1841 | 1.08 | 1775 | 1.12 | 1954 | 1.01 |
| A6-50-108 | 1676 | 1625 | 1.03 | 1652 | 1.01 | 1598 | 1.05 | 1746 | 0.96 |
| A6-30-80 | 1807 | 1635 | 1.11 | 1758 | 1.03 | 1757 | 1.03 | 1878 | 0.96 |
| A6-40-80 | 1996 | 1847 | 1.08 | 1914 | 1.04 | 1867 | 1.07 | 2039 | 0.98 |
| A6-60-80 | 2277 | 2120 | 1.07 | 2130 | 1.07 | 2026 | 1.12 | 2273 | 1.00 |
| A8-50-80 | 2139 | 2049 | 1.04 | 2189 | 0.98 | 2175 | 0.98 | 2343 | 0.91 |
| A10-50-80 | 2396 | 2063 | 1.16 | 2320 | 1.03 | 2378 | 1.01 | 2494 | 0.96 |
| A12-50-80 | 2600 | 2036 | 1.28 | 2416 | 1.08 | 2551 | 1.02 | 2612 | 1.00 |
| Mean | - | - | 1.10 | - | 1.04 | - | 1.06 | - | 0.97 |
| SD | - | - | 0.07 | - | 0.03 | - | 0.05 | - | 0.03 |
| Coefficient of variation | - | - | 0.06 | - | 0.03 | - | 0.04 | - | 0.03 |
| A6-50 | 2319 | 2226 | 1.04 | 2277 | 1.02 | 2174 | 1.07 | 2457 | 0.94 |
| A8-50 | 2681 | 2364 | 1.13 | 2556 | 1.05 | 2537 | 1.06 | 2769 | 0.97 |
| A10-50 | 2927 | 2434 | 1.22 | 2796 | 1.06 | 2878 | 1.03 | 3042 | 0.98 |
| A12-50 | 3376 | 2453 | 1.38 | 3003 | 1.13 | 3197 | 1.06 | 3286 | 1.03 |
| Mean | - | - | 1.20 | - | 1.06 | | 1.05 | - | 0.98 |
| SD | - | - | 0.14 | - | 0.05 | | 0.01 | - | 0.04 |
| Coefficient of variation | - | - | 0.12 | - | 0.04 | | 0.01 | - | 0.04 |

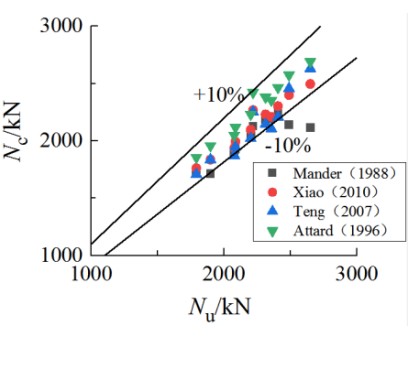

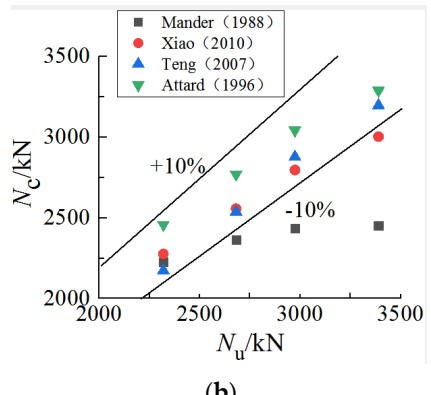

| (a) | (b) |
|---|---|

**Figure 17.** Comparison of axial compressive strength predictions. (**a**) ATCC-CHS, (**b**) ATCC.

*4.4. Economic Analysis*

Taking $L$ = 500 mm and $D_0$ = 6 mm as standards, the cost of unit specimen is shown in Table 6. Compared with CFAT-CHS, the price of ATCC-CHS only increases by 8.2% when the local buckling at the top of the aluminum tube is alleviated. Compared with STCC-CHS and STCC, the prices of ATCC-CHS and ATCC increased by 20.6% and 26.5% while alleviating the corrosion problem of steel tubes.

**Table 6.** Cost of the unit specimen.

| Specimen | ATCC-CHS | ATCC | CFAT-CHS | STCC-CHS | STCC |
|---|---|---|---|---|---|
| Cost of materials (yuan) | 600 | 550 | 620 | 420 | 370 |
| Cost of labor (yuan) | 400 | 260 | 300 | 400 | 260 |
| Cost of machinery (yuan) | 50 | 50 | 50 | 50 | 50 |
| total (yuan) | 1050 | 860 | 970 | 870 | 680 |

**5. Conclusions**

In this study, the performance of ATCC-CHS and ATCC in axial compression was investigated experimentally. The results support the following conclusions.

(1) The confinement coefficient is a major determinant of the failure modes of such columns. The specimens underwent brittle fracture for $\zeta \leq 1.35$, and localized buckling failure was observed for $\zeta > 1.35$. For thin-walled specimens, fiber reinforced polymer could be used to strengthen them.
(2) The load-bearing capacity of ATCC-CHS and ATCC can be increased by reducing the diameter–thickness ratio and the hollow rate or using stronger concrete. The thickness of the aluminum alloy tube was the most influential factor.
(3) Confinement of the outer aluminum tube played a significant role in the compressive strength of ATCC columns. When the thickness of the outer steel pipe increased from 6 to 12 mm, the axial compression stiffness of ATCC-CHS and ATCC increased by 6.5% and 116.2%, respectively.
(4) The $k$ increased with increases in the confinement coefficient $\zeta$, as did the *SLI*s of ATCC, while the *SLI*s of ATCC-CHS decreased with increases in $\zeta$. In practice, the cost and mechanical performance of an ATCC-CHS must be optimized jointly.
(5) The bearing capacity of an ATCC was 18.8% higher than that of a similar CFAT.
(6) The bearing capacity of a ATCC-CHS was 12.0% lower than that of a similar CFAT-CHS.
(7) The predictions of Teng's formula for the ultimate bearing capacity of an ATCC agree well with experimental observations.
(8) Attard's formula gives good predictions for ATCC-CHS.

Due to the large number of parameters, only one specimen was designed for each parameter in this paper. In future studies, a large number of parallel tests will be carried out as further verification. A useful next step would be to create non-linear finite element models of ATCC-CHSs and ATCCs to study the confinement mechanism and failure process, as well as to better quantify the influences of different parameters on their ultimate bearing capacity. The ultimate aim should be to create a calculation model which can serve as a reliable reference in engineering design.

STCCs are increasingly used in high-rise buildings and marine structures. An ATCC-CHS or ATCC made with aluminum rather than steel will deliver better durability even in a complex environment.

**Author Contributions:** Conceptualization, J.Z.; data curation, D.Z.; formal analysis, D.Z.; funding acquisition, L.L.; investigation, D.Z.; methodology, H.L.; project administration, J.Z.; resources, J.Z.; software, Z.M.; supervision, J.Z.; validation, D.Z.; visualization, D.Z.; writing—original draft, D.Z.; writing—review and editing, D.Z. All authors have read and agreed to the published version of the manuscript.

**Funding:** The authors disclosed receipt of the following financial support for the research, authorship, and/or publication of this article: this research was financially supported by the NSFC-Shandong Joint Fund (Grant Nos. U2106222), the National Natural Science Foundation of China (Grant Nos. 52108282), and the Natural Science Foundation of Shandong Province (ZR2021QE053).

**Institutional Review Board Statement:** Not applicable.

**Informed Consent Statement:** Not applicable.

**Data Availability Statement:** The data presented in this study are available on request from the corresponding author.

**Conflicts of Interest:** The authors declared no conflict of interest.

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
