# Peer review of "The Strength in Axial Compression of Aluminum Alloy Tube Confined Concrete Columns with a Circular Hollow Section: Experimental Results"

_buildings, doi:10.3390/buildings12050699_

Round 1

Reviewer 1 Report

Dear authors You have presented extensive and well-documented research. I have no significant substantive comments. I only have a few questions that require additional discussion or correction:

  • lines 151-153 - it can be commented that the yield point has been defined for 0.2% plastic strain,
  • point 3.1 (Failure modes) - you often use the term Nu, its physical meaning is defined in the next point (line 235), it should be defined with the first use,
  • point 3.2 - the discussed value is the confinement coefficient ζ, its meaning is defined in the footer of the Table 1 (line 144). I propose to move it to the body text. This coeffitient is one of the key values discussed in the manuscript and should be defined similarly to the k factor as a numbered equation.
  • point 3.3 - ε i missing in the description of N-ε curves (lines 342, 355 and 366)
  • standardize the way of presenting variables (L, Do, fcuof ...) in the text (I suggest writing them in italic with subscripts - as in equations)

Reviewer 2 Report

Article

The Strength in Axial Compression of Aluminum Alloy Tube Confined Concrete Columns with a Circular Hollow Section: Experimental Results

The article is well written.

A few suggestions are given below;

The results of the proposed formulas may include in the abstract as well.

"In this research, a new aluminum alloy tube confined concrete column with a circular section (termed an ATCC-CHS) was studied based on the CFDAT and STCC designs."

How about the cost of these tubes? authors need to justify the cost and include some comparison of cost with other materails.

If the authors have considered only one column for each parameter then there is a need to clearly state this in the article and also how about the results of the single column? usually, average results are considered for such small-scale columns.

Results are presented well, however, the results are not compared with the previous studies.

Figure 17, please include references as well.
